# Sampling Variation of RAD-Seq Data from Diploid and Tetraploid Potato (*Solanum tuberosum* L.)

**DOI:** 10.3390/plants10020319

**Published:** 2021-02-07

**Authors:** Zhenyu Dang, Jixuan Yang, Lin Wang, Qin Tao, Fengjun Zhang, Yuxin Zhang, Zewei Luo

**Affiliations:** 1Laboratory of Population and Quantitative Genetics, Institute of Biostatistics, Fudan University Shanghai, Shanghai 200433, China; 17210700056@fudan.edu.cn (Z.D.); 16110700059@fudan.edu.cn (J.Y.); wanglin@fudan.edu.cn (L.W.); 17110700109@fudan.edu.cn (Q.T.); 18110700015@fudan.edu.cn (Y.Z.); 2Qinghai Academy of Agricultural and Forestry Sciences, Xining 200433, China; 11110700065@fudan.edu.cn; 3School of Biosciences, University of Birmingham, Birmingham B15 2TT, UK

**Keywords:** sampling variation, overdispersion, RAD-seq data, *Solanum tuberosum* L.

## Abstract

The new sequencing technology enables identification of genome-wide sequence-based variants at a population level and a competitively low cost. The sequence variant-based molecular markers have motivated enormous interest in population and quantitative genetic analyses. Generation of the sequence data involves a sophisticated experimental process embedded with rich non-biological variation. Statistically, the sequencing process indeed involves sampling DNA fragments from an individual sequence. Adequate knowledge of sampling variation of the sequence data generation is one of the key statistical properties for any downstream analysis of the data and for implementing statistically appropriate methods. This paper reports a thorough investigation on modeling the sampling variation of the sequence data from the optimized RAD-seq (Restriction sit associated DNA sequencing) experiments with two parents and their offspring of diploid and autotetraploid potato (*Solanum tuberosum* L.). The analysis shows significant dispersion in sampling variation of the sequence data over that expected under multinomial distribution as widely assumed in the literature and provides statistical methods for modeling the variation and calculating the model parameters, which may be easily implemented in real sequence datasets. The optimized design of RAD-seq experiments enabled effective control of presentation of undesirable chloroplast DNA and RNA genes in the sequence data generated.

## 1. Introduction

Development of next-generation sequencing technology (NGS) has enabled the identification of sequence variant-based genetic molecular markers at a genome-wide scale, a population level, and a very competitive cost in comparison to traditional DNA molecular markers such as restriction fragment length polymorphisms (RFLPs), amplified fragment length polymorphisms (AFLPs), and single-nucleotide polymorphisms (SNPs) [1,2]. This has motivated great interest in genotyping by sequencing (GBS) for population and quantitative genetic analyses in diploid and tetraploid species [3]. It is established that the use of genotype information at molecular markers may significantly improve the efficiency of genetic analysis, particularly in tetraploids [4].

GBS is relatively straightforward in diploid species, although serious consideration must be given to several major sources of variation in collecting and processing the sequencing data for accurate identification of allele-specific sequencing reads [5]. GBS in tetraploids is a much more challenging task and involves distinguishing the number of each constituent allele (i.e., the allele dosage) in a heterozygote genotype (i.e., Uitdewilligen [6]). However, the reliability and accuracy of NGS heavily rely on knowledge of the nature of variation embedded in the sequence data. The variation may be biological or nonbiological in nature, and it may be associated with technical issues such as errors associated in process of sequencing library construction, sequencing errors, and errors stemmed from data processing [5,7,8,9].

Tremendous research has been focused on modeling the complexities in variation pattern and structure of diploid sequencing data [10,11]. Sequence data generated from polyploids such as cultivated potato (*Solanum tuberosum* L.) show much more sophisticated variation than diploid sequence data. In diploids, homozygote and heterozygote genotypes at a polymorphic site can be inferred directly from sequence data, and GBS in diploids is, thus, relatively trivial. However, GBS in polyploids represents a much more challenging task; for example, there would be five possible genotypes at a biallelic site (A and a) of a tetraploid genome, i.e., AAAA, AAAa, AAaa, AAAa, and aaaa. The heterozygote genotypes (A_a_) are indistinguishable from each other using sequence data. Coupled with other sources of errors, polyploid sequence data were recognized as being “messy” for their complicated sampling distribution in Gerard et al. [12]. Gerard et al. made a comprehensive survey of the impacts of several key sources of variation in hexaploid sweet potato (*Ipomoea batatas*) sequence data for GBS [12]. Among the variation sources discussed in the literature, sampling variation is the ultimate and key statistical property of sequencing data, and it is essential information for the reliability of modeling and any downstream analysis with the data. They pointed out that the “messy” hexaploid sequence data may involve dispersion over standard independent distributions, but little is known about to what extent the data deviate from a specific distribution and what form of the statistical distribution the data follow.

This paper represents statistical methods for modeling sampling variation of new-generation genomic sequence data from diploid and tetraploid plants and for estimating the model parameters from the sequence data. These methods were demonstrated through analyzing the RAD-seq (Restriction site associated DNA sequencing) [13] data from diploid and tetraploid parental lines and their offspring individuals of potato (*Solanum tuberosum* L.). Lastly, we discussed how the sampling variation pattern predicted from the analysis may influence quantitative genetic analysis involving use of the next-generation genomic sequence data.

## 2. Results

### 2.1. Sequence Data Collected

We collected sequence read data from two pooled sequence libraries for diploids and tetraploids of *Solanum tuberosum* L., each of which comprised 12 samples (two parental lines and 10 offspring). The diploid and tetraploid potato strains used to generate the offspring populations are detailed in Section 4. The designed length of DNA segments targeted in the RAD-seq experiment varied between 360 and 560 bps. After chopping the adapter and PCR primer sequences of 136 bps, the actual selected DNA segments were in the range of 224–424 bps as demonstrated in Figure 1a,b, with the mean lengths of the DNA segments being 317 bp and 310 bp, respectively. Figure 1c,d show the number of reads in each of the pooled RAD-seq libraries of diploid or tetraploid potato, which was approximately equal to 4 M, i.e., the designed number of sequence reads for each of the samples, demonstrating the uniform presentation of the component samples in the pooled RAD-seq libraries. These findings show that the designed parameters of the RAD-seq library construction were well met and realized.

### 2.2. The Efficiency of the RAD-Seq Protocol to Remove the Chloroplast and Ribosomal RNA (rRNA) DNA Fragments

Raw short reads after the quality check were aligned to the potato reference genome using Bowtie2 [14] according to the mapping quality criteria set in Section 4. When the reads were collected from the library without designed removal of DNA from the chloroplast and rRNA genes, we showed that fewer than one-third of the sequence reads were aligned to the genomic sequence in diploid (27%) and tetraploid (30%) genomes (Table 1). In contrast, when chloroplast and rRNA sequences were designed to be removed by implementing a second round of digestion, the majority of reads were successfully mapped to the reference genomic sequence in the diploids (86%) or tetraploids (85%) of potato (Table 1). Only small proportions of the sequence reads (5–7%) were mapped to the chloroplast genomes and the rRNA genes. These results indicate that the design objectives of the optimized RAD-seq approach were successfully achieved in effectively minimizing the presentation of the chloroplast and rRNA in the RAD-seq libraries and in significantly increasing the proportion of reads mapped to the reference genomic sequence.

### 2.3. Preliminary Bioinformatic Analysis of the RAD-Seq Data

The RAD-seq data collected from this study were used to fit the two alternative sampling distributions (binomial distribution and *β*-binomial distribution). However, sequence coverage and polymorphic segregating alleles may vary considerably from one polymorphic site to the other in the RAD-seq dataset. To minimize these influences, we further screened the RAD-seq data to be included into the model fitting on the basis of the following screening and grouping criteria: the selected data for the modeling fitting must carry a polymorphic site with at least two alleles in the diploid or tetraploid samples and have a coverage of ≥20. The selected sequence data were then grouped according to their coverage into [20,60), [60,100). Within each of the groups, we assigned one of the polymorphic nucleotides (usually the one from the reference genome) as allele *A* and the other as *a*, and the number of *A*-carrying sequence reads was counted as n_A_. The number of the polymorphic sites in each of the groups was denoted by *M*.

According to the above criteria, we were able to identify a total of 59,503 biallelic sites between the two diploid parents and a total of 68,389 biallelic sites between the two tetraploid potato parents. Among them, there were 28,984 or 31,879 sites common between the two diploid or tetraploid parents. Use of FreeBayes [15] software enabled genotyping at these polymorphic sites in both the diploid and the tetraploid groups, as tabulated in Table 2.

The above-predicted genotypes at the selected polymorphic sites were used in the subsequent model fitting analysis.

### 2.4. Sampling Distribution Fitting

We fitted the RAD sequence data at the identified biallelic nucleotide sites, which accounted for 95% of the RAD sequence scanned, from the above diploid and tetraploid parents and their offspring individuals to the two candidate distributions, binomial and *β*-binomial distributions. For illustrative purposes, we showed the expected number of allele *A* (the others were labeled a) from the candidate distributions and compared it to the observed number. Figure 2a,b show frequencies of the observed and expected numbers of the reference allele under the candidate distributions from RAD-seq data from all diploid and tetraploid individuals listed in Table 2 when the coverage of polymorphic sites was between 20 and 60. Figure 2c,d show frequencies of the observed and expected numbers of the reference allele when the sequence coverage of polymorphic sites was between 60 and 100. To test for goodness of fit between the observed and expected numbers of allele *A*, we calculated ${\hat{\chi }}_{df}^{2}$, and we present the ratio of ${\hat{\chi }}_{df}^{2}/df$ in Figure 3a,b for the sequence coverages 20–60 and 60–100, respectively.

It is clear from Figure 2 and Figure 3 that the sampling variation of the RAD-seq data was clearly and substantially better modeled by the *β*-binomial distribution than by the binomial distribution in both diploid and tetraploid sequence data.

## 3. Discussion

Advancement in new-generation sequencing techniques has stimulated a wide spectrum of analyses in modern genetics and genomics. The sampling distribution of the sequence data generated from the techniques is one of the most important features of the data, and a good understanding of this statistical property is essential for sequence data to be appropriately implemented into relevant analyses. For example, a binomial distribution has been widely assumed in prediction of genotypes at polymorphic sites called from sequence data in both diploids [5,10,16,17] and polyploids [2,18,19]. Gerard et al. demonstrated that the sampling variation of real sequence data deviates substantially from that under bi- or multinomial distributions, although these authors did not provide a further investigation into how the dispersed verion would be statistically approriately modeled [12].

Generation of sequence data can be assimilated to a random process of sampling a number of alleles carried by an individual genotype at any given site. This process may be subject to a wide range of technical and biological variations, as thoroughly reviewed in the literature. Statistically, binomial (or multinomial) distribution models a random process of independently and probablistically identical sampling from two (bi-) or multiple objects. The present study demonstrates that the RAD-seq data collected from the present study showed markedly wider variation than that expected under binomial distribution, whilst the *β*-binomial fit the data variation much better than the binomial distribution.

The i.i.d (identical and independent distribution) assumption behind the bi- or multinomial distribution may rarely be satisfied in the sampling process of generation of any sequence data. For instance, different primer and/or template sequences may be subjected to marked variation of PCR products in sequence library construction [20]. The efficiency in sysnthesis of sequence reads depends on the concentration and sequence of the template pool [21]. The inheritent features in the process of sequence data generation and errors involved in every step of the bioinformatic process of sequence data may substantially violate the i.i.d assumption; thus, binomial or multinomial distributions cannot be recognized to be a statistically appropriate model for smapling variation of the sequence data, particularly the data located at the distribution tails of the data, as shown in the present study.

The deviation in sampling variation of sequence data from that of bi- or multinomial distribution, as demonstrated in the present study, would have significant impacts on and bias the downstream analyses. For instance, when the sequence data are used to predict genotypes at the sequence variant sites, the probabilities of the predicted genotypes will be severely biased from the sequence reads which are at tails of the sequence data distribution, as shown in Figure 2. Although use of the predicted genotypes has been demonstrated to improve the efficiency of quantitative genetic analyses in both diploids and tetraploids through computer simulation studies [5,22,23], little is known about the impacts of biased genotype prediction on these analyses. Obviously, an adequate knowledge of sampling distribution of sequence data represents the prerequisite for the reliability of sequence-based genotyping and, in turn, the reliability of any analysis based on the genotyping information. The present study revealed a key feature of sequence data and highlighted the importance of an essential step in genetic and genomic analyses using new-generation sequence data, as well as provided methods for fitting new-generation sequence data to a *β*-binomial distribution and estimating the corresponding model parameters.

The present study implemented the optimized RAD-seq experiments for sequencing parental varieties and the first-generation offspring of diploid and tetraploid potatoes (*Solanum tuberosum* L.). The RAD-seq experiments enabled an adequate length selection of DNA segments that were designed for an even coverage of the target genome, minimizing representation of chloroplast DNA and RNA genes in the sequence library and, in turn, maximizing gain of the target sequence data.

## 4. Materials and Methods

### 4.1. Creation of Diploid and Tetraploid Segregation Populations of Solanum tuberosum L.

We created two segregation offspring populations from crossing two highly heterozygous diploid potato strains (BD6-6 and BD66-6) or two tetraploid potato cultivars (Atlantic and Longsu-3). These parental strains vary significantly in a series of morphological and developmental traits and were provided by Crop Institute of Qinghai Academy of Agriculture and Forestry Sciences (Qinghai, China) where the cross-breeding and field experiments were conducted. Although there were a total of 184 diploid and 301 tetraploid offspring together with their parental lines successfully collected from the crossing experiments, in the present study, only 10 offspring individuals and their parents were implemented from each of the two outbred segregation populations. Selection of these offspring individual samples was largely random for demonstrative purposes. Leaf samples were collected when the plants bloomed the first flower, and 10–20 g of fresh leaves were collected for each of the plants.

### 4.2. Construction of RAD-Seq Libraries

DNA samples were first extracted from the leaf samples of the selected individual plants as described above using the DNeasy Plant Mini Kit (QIAGEN, Valencia, CA, USA) to extract DNA, and the sequence libraries of the selected DNA sampled were constructed following the method we previously described in [24]. The sequence library construction protocol was modified in two aspects. Specifically, DNA segments with target length were selected in two steps, firstly by the Pippin prep system, and secondly further refined by use of Ampure XP beads. This effectively improved the accuracy of selection for DNA segments with the designed fragment length. The workflow and protocol of the RAD-seq library construction are diagrammatically illustrated in Figure 4. Adaptors used in the library construction are listed in Appendix A.

The constructed RAD-seq libraries of 12 samples were pooled into an integrate library to be sequenced by an Illumine High-2000 sequencer to generate an average of 4 M reads of 2 × 150 bps for each of the 24 biological samples. We stress that the RAD-seq protocol implemented here is an optimized RAD-seq approach that minimizes presentation of untargeted DNA segments from chloroplast DNA and RNA genes, as detailed in our previous work [24].

### 4.3. Preliminary Processing of the Sequence Data

The RAD-seq data collected were firstly checked for quality and filtered for the next step of analysis. The sequence reads were removed from further analyses if they had an average Phred score below 20, which was assessed by use of the software trim-galore, or mapping quality lower than 20, which was worked out by using the software Bowtie2. Moreover, the paired reads mapped more than 500 bps apart were excluded from further analyses. The potato reference genome was used for the quality screening analysis and was downloaded from http://potatogenomics.plantbiology.msu.edu.

### 4.4. Identifying SNPs from the Sequence Data

The sequence reads after the above quality filtering process and with a mapping coverage greater than 20 were subjected to screening for single-nucleotide polymorphisms embedded in the sequence reads. A nucleotide site is called polymorphic if there are two (diploids) or more (tetraploids) nucleotides present at the site. We removed those variants with <5% of the reads to the improve statistical efficiency of the subsequent analyses.

### 4.5. Calling Polymorphic Sites and Genotype at the Identified Sites

It is straightforward to determine a diploid individual genotype at a polymorphic site within sequence reads. However, there would be three possible genotypes at a biallelic or triallelic site for a tetraploid heterozygote; thus, it is not trivial to predict tetraploid genotypes even from sequence data [6,25]. We implemented the method “freebayes” described in Garrison [15] to predict tetraploid genotypes of the tetraploid individuals from their sequence data. The method predicts the probability of a sample genotype at a heterozygous locus given sequence data through an approximation Bayes formula. The method was designed to model short-read sequence data of independent samples. It predicts both polymorphic sites and genotypes at the sites using a computationally efficient algorithm through a series of computationally tractable approximation algorithms, particularly when the number of individuals and the number of polymorphic sites are large.

### 4.6. Sampling Distributions of Sequence Data

For a given individual with a ploidy level k (=2 or 4), its genotype is denoted by $A^{k_{A}}C^{k_{C}}G^{k_{G}}T^{k_{T}}$,with $k_{X}$ being the number of allele *X* = *A*, *C*, *G*, or *T* and $k_{A}+k_{C}+k_{G}+k_{T}=$2 (diploids) or 4 (tetraploids). The individual is observed in the RAD-seq experiment to have n_X_ sequence reads carrying *X* = *A*, *C*, *G*, and *T*. Sampling variation of the RAD-seq is characterized by the following conditional probability distribution: $\mathrm{Pr}\{n_{A},n_{C},n_{G},n_{T}|k_{A},k_{C},k_{G},k_{T}\}$. We explore here several cases of patterns of sampling variation of the RAD-seq data, i.e., the form of the probability distribution. When the genotype allele is independently sampled in the process of sequencing, $n_{A},n_{C},n_{G}\;\mathrm{and}\;n_{T}$ follow a multinomial distribution with the form given below
(1)$$\mathrm{Pr}\{n_{A},n_{C},n_{G},n_{T}|k_{A},k_{C},k_{G},k_{T}\}=\left(\begin{array}{c} n\\ n_{A}n_{C}n_{G}n_{T} \end{array}\right)\prod_{X}^{\left(A,C,G,T\right)}{\left(k_{X}/k\right)}^{n_{X}},$$
where $n=n_{A}+n_{C}+n_{G}+n_{T}$ and $k=k_{A}+k_{C}+k_{G}+k_{T}$. Equation (1) indicates an ideal circumstance, i.e., sampling of alleles in an individual genotype is independent in the process of sequence library construction, sequencing, and later sequence data processing. This independence assumption has been widely made in the recent literature [2,3,7,10]. The mean and variance of the multinomial distribution are $n\prod_{X}^{\left(A,C,G,T\right)}\left(k_{X}/k\right)$ and $n\prod_{X}^{\left(A,C,G,T\right)}\left(k_{X}/k\right)\left(1-k_{X}/k\right)$.

However, many empirical analyses have demonstrated severe deviation of sequence data from this independence assumption [1,10,12]. We proposed here the multivariate Polya distribution [26] as a more general form to model the sampling distribution of $n_{A},n_{C},n_{G}\;\mathrm{and}\;n_{T}$ in the present context of sequence data analysis. The Polya distribution is a compound probability distribution of a general multinomial distribution with Bernoulli trial probability parameters $\alpha_{X}$ (*X* = *A*, *C*, *G*, *T*) being sampled from the Dirichlet multinomial distribution, as given by
(2)$$\mathrm{Pr}\{n_{A},n_{C},n_{G},n_{T}|k_{A},k_{C},k_{G},k_{T}\}=\frac{\left(n!\right)\Gamma \left(\sum_{X}^{\left(A,C,G,T\right)}\alpha_{X}\right)}{\Gamma \left(n+\sum_{X}^{\left(A,C,G,T\right)}\alpha_{X}\right)}\prod_{X}^{\left(A,C,G,T\right)}\frac{\Gamma \left(n_{X}+\alpha_{X}\right)}{n_{X}!\Gamma \left(\alpha_{X}\right)}.$$

When Equation (2) is conjugated with Equation (1), the marginal probability distribution of $n_{A},n_{C},n_{G}\;\mathrm{and}\;n_{T}$ is given by
(3)$$\mathrm{Pr}\{n_{A},n_{C},n_{G},n_{T}|\alpha_{A},\alpha_{C},\alpha_{G},\alpha_{T}\}=\frac{nB\left(\alpha_{S},n\right)}{\prod_{X,n_{X}>0}^{\left(A,C,G,T\right)}B\left(\alpha_{X},n_{X}\right)},$$
where $\alpha_{S}=\alpha_{A}+\alpha_{C}+\alpha_{G}+\alpha_{T}$ and the beta function B(x,y)=Γ(x)Γ(y)/Γ(x+y). Equation (3) can model a much wider spectrum of variation, i.e., overdispersion, in sampling the sequence data, and it is appropriate for sequence data from a species of any ploidy levels. Although there is no technical problem when developing statistical analysis of the sequence data with the probability model (Equation (3)) for other numbers of segregating alleles at a polymorphic site, we focused here on diploid and tetraploid sequence data only. In diploids, each individual has up to two alleles at each SNP site. In principle, there may be up to four alleles at an SNP site in tetraploids. However, empirical surveys show that biallelic SNPs have accounted for ~96% of polymorphic sites identified from tetraploid potato sequence data [6] (Uitdewilligen et al. 2013; Luo et al. unpublished data). Approximately 95% of biallelic sites were observed in the dataset analyzed in the present stduy. Thus, we focused here on the biallelic case for both diploid and tetraploid sequence datasets. Without loss of generality, we denoted the two alleles *A* and *a*. Equations (1) and (3) could be simplified into
(4)$$\mathrm{Pr}\{n_{A}|n,k_{A}\}=\left(\begin{array}{c} n\\ n_{A} \end{array}\right){\left(k_{A}/k\right)}^{n_{A}}{\left((n-k_{A})/k\right)}^{n-n_{A}},$$
which is the probability function of binomial distribution with mean and variance $n\times k_{A}/k$ and $n\times k_{A}\left(k-k_{A}\right)/k^{2}$, and, in general,
(5)$$\mathrm{Pr}\{n_{A}|n,\alpha_{A},\alpha_{a}\}=\left(\begin{array}{c} n\\ n_{A} \end{array}\right)\frac{B\left(n_{A}+\alpha_{A},n-n_{A}+\alpha_{a}\right)}{B\left(\alpha_{A},\alpha_{a}\right)}=\left(\begin{array}{c} n\\ n_{A} \end{array}\right)\frac{\Gamma \left(\alpha_{A}+\alpha_{a}\right)\Gamma \left(n-n_{A}+\alpha_{a}\right)\Gamma \left(\alpha_{A}+\alpha_{a}\right)}{\Gamma \left(n+\alpha_{A}+\alpha_{a}\right)\Gamma \left(\alpha_{A}\right)\Gamma \left(\alpha_{a}\right)},$$
which is the probability mass function of beta binomial distribution with mean and variance $n\alpha_{A}/(\alpha_{A}+\alpha_{a})$ and $n\alpha_{A}\alpha_{a}(\alpha_{A}+\alpha_{a}+n)/[{\left(\alpha_{A}+\alpha_{a}\right)}^{2}(\alpha_{A}+\alpha_{a}+1)]$. Equation (5) involves a series of gamma functions Γ(z), and their numerical calculation would be computationally tedious, particularly for a large value of z. Yang proposed an approximation of gamma functions, as given below [27].
(6)$$\Gamma (z)=\Gamma (y+1)≅\sqrt{2\pi y}{\left(\frac{y}{e}\right)}^{y}{\left(y\sinh\frac{1}{y}\right)}^{y/2}\exp(\frac{7}{324}\frac{1}{y^{3}(35y^{2}+33)}).$$

Accuracy of the approximation is on the order of 10^−4^ when z→∞. The first and second moments of the beta binomial distribution can be calculated from
(7)$$\mu_{1}=E(n_{A})=\frac{n\alpha_{A}}{\alpha_{A}+\alpha_{a}},$$
(8)$$\mu_{2}=E(n_{A}^{2})=\frac{n\alpha_{A}[n(1+\alpha_{A})+\alpha_{a}]}{\left(\alpha_{A}+\alpha_{a})(1+\alpha_{A}+\alpha_{a}\right)}.$$

Setting them as equal to estimate ${\hat{\mu }}_{1}=\sum_{i}^{M}n_{Ai}/M$ and ${\hat{\mu }}_{2}=\sum_{i}^{M}n_{Ai}^{2}/M$ from a sample of $n_{A1}$, $n_{A2}$, …, $n_{AM}$, we can calculate the model parameters $\alpha_{A}$ and $\alpha_{a}$ from
(9)$${\hat{\alpha }}_{A}=\frac{n{\hat{\mu }}_{1}-{\hat{\mu }}_{2}}{n\left({\hat{\mu }}_{2}/{\hat{\mu }}_{1}-{\hat{\mu }}_{1}-1\right)+{\hat{\mu }}_{1}},$$
(10)$${\hat{\alpha }}_{a}=\frac{\left(n-{\hat{\mu }}_{1})(n-{\hat{\mu }}_{2}/{\hat{\mu }}_{1}\right)}{n\left({\hat{\mu }}_{2}/{\hat{\mu }}_{1}-{\hat{\mu }}_{1}-1\right)+{\hat{\mu }}_{1}}.$$

Parameters characterizing the above three possible sampling distributions can be calculated from the sample data. Using these parameter estimates and the corresponding probability distribution function (Equation (5)), one can calculate the expected value for each $n_{Ai}$ as ${\tilde{n}}_{Ai}$ (*i* = 1, 2, …, M), and we conducted a goodness-of-fit test between the expected and observed $n_{Ai}$ through an empirical chi-square test. An estimate of the test statistic is calculated by
(11)$${\hat{\chi }}_{df=M-1}^{2}={\sum_{i=1}^{M}\left(n_{Ai}-{\tilde{n}}_{Ai}\right)}^{2}/{\tilde{n}}_{Ai}^{2},$$
with df=M−1 degrees of freedom. Significance of the goodness-of-fit test is characterized by the *p*-value, which is calculated from
(12)$$P=\mathrm{Pr}\left\{\chi_{df=M-1}^{2}>{\hat{\chi }}_{df=M-1}^{2}\right\}=1-\mathrm{Pr}\left\{\chi_{df=M-1}^{2}\leq {\hat{\chi }}_{df=M-1}^{2}\right\},$$
in which $\chi_{df=M-1}^{2}$ is the chi-square variable with df=M−1 degrees of freedom.

## Figures and Tables

**Figure 1 plants-10-00319-f001:**
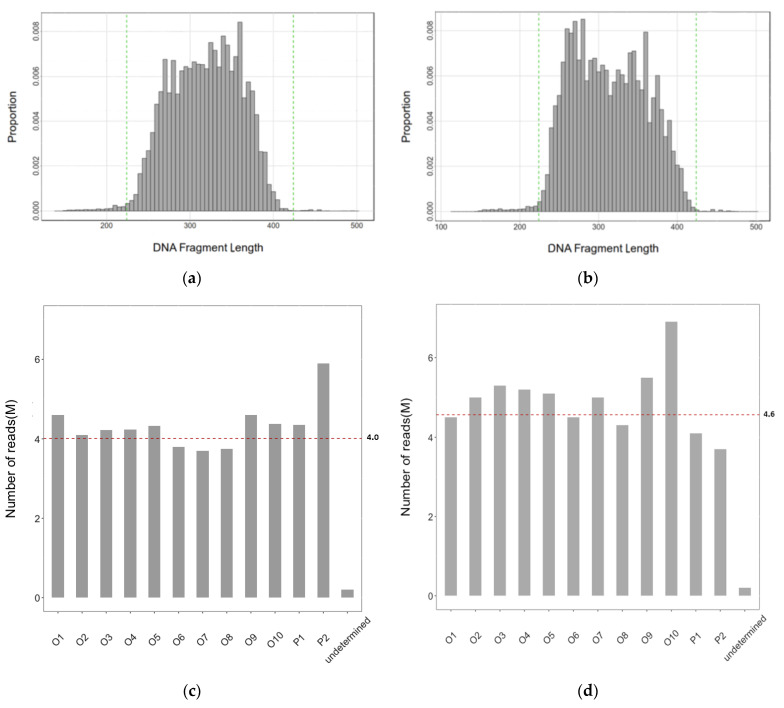
Distribution of the lengths of DNA segments (**a**,**b**) and the number of sequence reads in each of the pooled RAD-seq libraries comprising 12 diploid and tetraploid samples (**c**,**d**). The green lines in (**a**,**b**) bracket the ranges of the designed length of the DNA segments. The red dashed lines in (**c**,**d**) show the average number of reads per sample. (**a**,**c**) Diploids; (**b**,**d**) tetraploids.

**Figure 2 plants-10-00319-f002:**
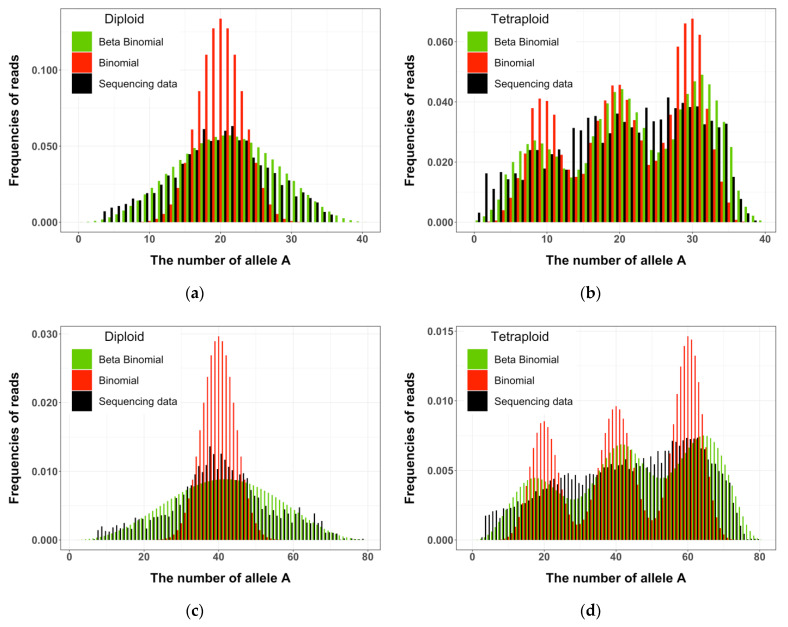
The histograms of the observed and expected numbers of the reference allele from the sequencing data of diploid and tetraploid potato at the coverage of 20–60 (**a**,**b**) or 60–100 (**c**,**d**). The expected values were calculated from binomial and *β*-binomial distributions.

**Figure 3 plants-10-00319-f003:**
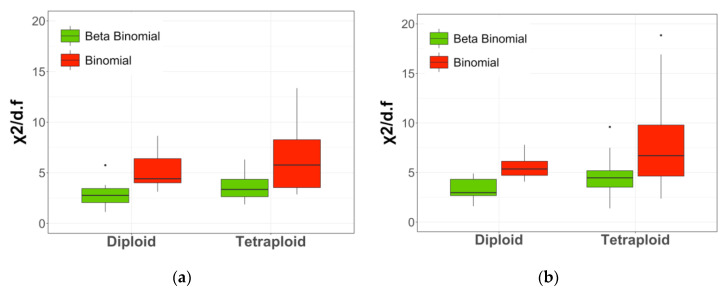
The boxplot of ${\hat{\chi }}_{df}^{2}/df$ for the goodness-of-fit test between the observed and expected numbers of sequence reads under two alternative distributions at two coverages (20–60 on the left (**a**) and 60–100 on the right (**b**)) from diploid and tetraploid potato.

**Figure 4 plants-10-00319-f004:**
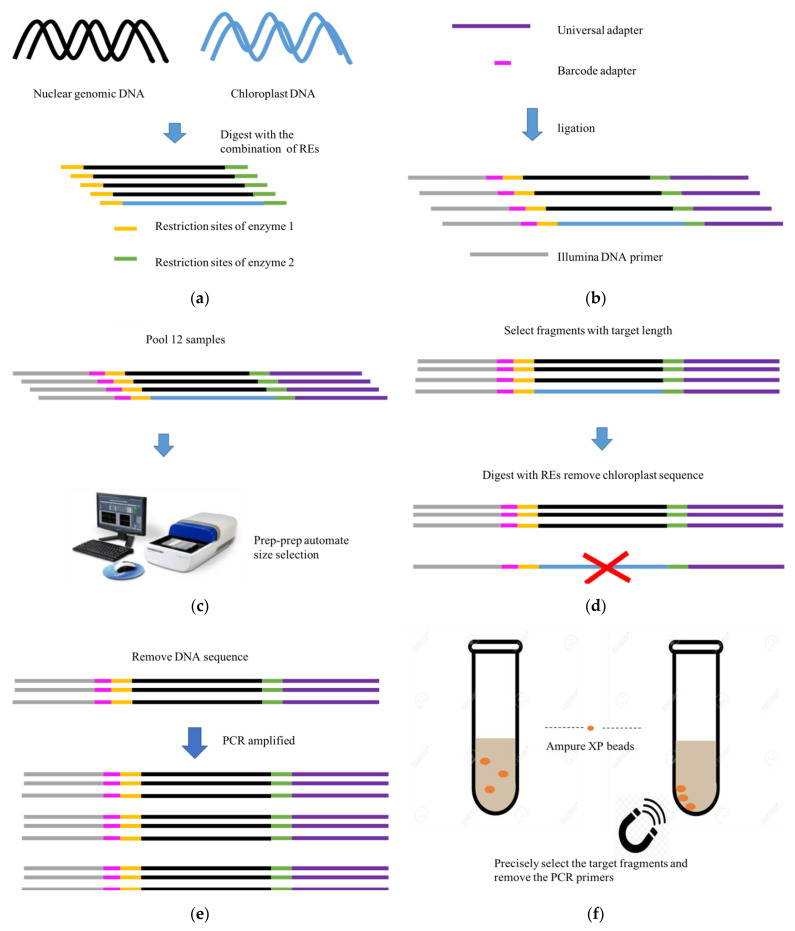
Diagrammatic workflow of the optimizing RAD-seq library construction in the present study. (**a**) Digesting genomic DNA into DNA fragments with designed lengths. (**b**) Adding adapters on both sides of the selected DNA fragments. (**c**) The first round of fragment size selection with Pippin prep. (**d**) The second digestion to remove DNA fragments from the chloroplast genome and/or RNA genes. (**e**) PCR amplification. (**f**) The second round of fragment size selection with Ampure XP beads.

**Table 1 plants-10-00319-t001:** Proportions (%) of the sequence reads aligned to different regions in the diploid or tetraploid potato genomes from the RAD-seq experiment. rRNA, ribosomal RNA.

Mapped Regions	Without Removing Chloroplast and rRNA Fragments		With Removing Chloroplast and rRNA Fragments	
	Diploid	Tetraploid	Diploid	Tetraploid
Genomic DNA	27.0	30.3	85.5	84.8
Chloroplast DNA	64.5	61.1	6.5	4.4
rRNA genes	0.7	1.2	0.3	0.3
Unmapped	7.8	7.4	7.7	10.5

**Table 2 plants-10-00319-t002:** The number of polymorphic markers screened from the RAD-seq datasets of diploid and tetraploid parental strains (P1 and P2) and 10 offspring individuals (O1, O2, …, O10).

Individuals	Diploids			Tetraploids				
	AA	Aa	aa	AAAA	AAAa	AAaa	Aaaa	aaaa
P1	6369	16,109	20,837	6355	12,420	7389	4776	17,905
P2	6314	12,992	25,866	6104	12,129	7804	5232	20,150
O1	6190	9712	15,781	6330	11,007	6747	5122	20,605
O2	5756	8471	16,875	5719	9556	6294	4549	18,086
O3	5657	8024	16,292	8779	13,662	8297	6727	24,261
O4	5843	10,034	15,295	6398	9618	6664	4131	21,851
O5	5812	9257	15,803	6609	11,194	7071	5152	21,951
O6	5181	5843	15,410	6508	10,303	6886	5137	19,245
O7	4904	8329	17,343	6965	10,571	7877	5145	20,327
O8	5294	10,134	19,844	6149	9854	6936	4444	18,988
O9	5562	10,918	23,296	5692	9535	6300	3968	15,634
O10	5450	7459	18,270	6999	12,306	7714	5269	21,468

## Data Availability

The data used in the paper may be obtained upon a request from the corresponding author.

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
