# Peer review of "Sampling Variation of RAD-Seq Data from Diploid and Tetraploid Potato (Solanum tuberosum L.)"

_plants, 2021, doi:10.3390/plants10020319_

Round 1

Reviewer 1 Report

The manuscript consists on optimization of RAD-seq protocol to be used in parental lines and offspring of diploid and tetraploid potato populations. The authors should improve the manuscript in general. The introduction does not reflect the importance of this study to be done in potato, why is it useful for the scientific community? The authors should explain better how the other scientists can implement the statistical model to calculate sampling variation, is there a pipeline or software? Where?

A previous study (Jiang et al 2016) presented a optimized RAD-seq method tested in diploid and tetraploid Arabidopsis and Solanum tuberosum, which is the novelty of this study? I strongly suggest that the authors should provide more discussion in the introduction mentioning the importance of this study and what’s new or what have improve from previous study (Jiang et al 2016).

Major comments

In the introduction the authors should be much clear about their objectives. They should explain which populations are generating and the reason to perform this analysis in potato.

In section 2.1, explain briefly which samples have been sequenced, which are the parental lines, where

Figure 1. The authors should indicate which graph corresponds to the diploid pool and which one to the tetraploid pool. Which is the mean read length for the diploid pool and the tetraploid pool? It could be useful to include

Figure 1a and 1b indicate the mean length of the DNA fragments.

Table 1. Indicate in the table that the values are percentage, use the symbol (%), example: “Reads mapped to (%)”

In section 4.1, the authors should explain better in which conditions they generate the two populations (temperature, replicates…) and how they select the 10 offspring individuals. Also, provide more information about the parental lines, which are the characteristics traits for each parental line? Why the authors have chosen these lines?

In section 4.2, how old are the leaves? Indicate the number of leaves per sample that was selected.

Figure 4 e. what “remove DNA sequence” means? It should be more appropriate “DNA-pooled sequences”

In section 4.3, please provide the name and the source of the standard pipeline used to process the RAD-seq data.

Minor comments

Scientific species name should be written in italics and first letter of genus in capitals, example: Solanum tuberosum.

Reviewer 2 Report

Dear authors,

I hope you are fine!

I read your manuscript and I only have some minor comments.

  1. Please, review the nomenclature of the species 'Solanum tuberosum' (title, abstract, introduction, and so on). Moreover, you should include the species name in Material and Methods.
  2. Use 'RAD-seq' through the manuscript.
  3. Subtitle '2.2 The efficiency to remove...' is not well understood.
  4. 'to the criteria in Method'. Whay method? Perhaps, you should modify the full sentence. 
  5. 'In contract, when chloroplast and rRNA...' by 'In contrast,...'?
  6. Table 1. Please, include '%' symbol in the table for a better understanding.
  7. When you use allele A and a, I think that this should be highlighted as 'A' and 'a' or something like that.
  8. Figure 1. What is the difference between panel (a) and (b)? and between (c) and (d)? diploids and tetraploids?. These differences should be indicated in the caption.
  9. Discussion. Some references are lacking. For instance,  in the final sentences of paragraph 1, and two sentences of paragraph 2. 
  10. Material and methods.
    1. I have doubts about what individuals used in this study are diploids or tetraploids. P1 and P2 are diploids or tetraploids? This fact should be clarified in material and methods as in the caption of Figura 1. Moreover, define 'P' and 'O' as Parental and Offspring, respectively, for a better understanding of readers. 
    2. Please, include the manufacturer of DNeasy Plant Mini Kit DNA.
    3. Cite 'Jiang{24}. I think that you should use only the reference number. 
    4. 'Primers used in the...'. Please, use 'adaptors' term.

Reviewer 3 Report

The authors investigated the modelling of the sampling variation of sequence data based on optimized RAD-seq experiments of diploid and autotetraploid potatoes. The results showed significant dispersion in sampling variation of the sequence data. The authors provided statistical methods for modeling the sampling variation and calculating the model parameters, which could be used by real sequence datasets. I believe that the authors have provided sufficient background, well explained the methodology used, presented the results appropriately, and withdrawn conclusions based on available data. I have a major technical concern about the presentation of its results and some minor editorial and grammatical suggestions throughout the entire manuscript with a few of them listed here for your consideration:

Major:

In section 2.3, please consider change the format of using bullets but explain these criteria in a normal text format. Also, the fourth bullet is unclear, particularly the use of letters without explanations is confusing.

Minor:

Use upper case of the first letter of the word “solanum” in title.

Figure 1, green lines (a-b) need explanations in the caption.

Table 1, insert “(%)” after “Proportions” in the title or somewhere in the table. Change “Reads Mapped to” to “Mapped regions” in the first column of Table 1.

Section 4.2., second paragraph, please correct “Jiang[24]”

Table 2, please explain that P1 and P2 are two parents, while O1-O10 represents 10 individuals of offspring.

Figure 2, please correct the grammatical errors in the caption of Figure 2.

Throughout the entire manuscript, please keep the consistency of:

“et al.” or “et al”

”RAD-seq” or ”RAD seq”

Round 2

Reviewer 3 Report

I appreciate very much the efforts that the authors have devoted to improving this manuscript. I have no further questions about this ms except for one minor issue: the "S" in the word "Solanum" of the title needs to be italicized.